# Molecular, Chemical, and Sensory Attributes Fingerprinting of Self-Induced Anaerobic Fermented Coffees from Different Altitudes and Processing Methods

**DOI:** 10.3390/foods11243945

**Published:** 2022-12-07

**Authors:** Silvia Juliana Martinez, Nádia Nara Batista, Ana Paula Pereira Bressani, Disney Ribeiro Dias, Rosane Freitas Schwan

**Affiliations:** 1Biology Department, Federal University of Lavras (UFLA), Lavras 37203-202, MG, Brazil; 2Food Science Department, Federal University of Lavras (UFLA), Lavras 37203-202, MG, Brazil

**Keywords:** natural and pulped natural, SIAF, DGGE, volatiles, coffee quality

## Abstract

Coffee quality is achieved by performing good practices. This study aimed to evaluate coffees from different altitudes fermented with the self-induced anaerobic method (SIAF) and processed via natural (N) and pulped natural (PN). Molecular (PCR-DGGE), chemical (HPLC, ABTS, DPPH, ATR-FTIR, and GC-MS), and sensory analyses were performed. *Leuconostoc* predominated both processes and all altitudes. *Hanseniaspora* and *Pichia* predominated both processes at 800 and 1200 m. Acids were higher in N coffees for all altitudes. Acetic, malic acid and alcohols were the most abundant. Higher sensory scores were obtained in N (mainly at 1400 m—88.13). Floral and spices were perceived in all samples. ABTS capacity in roasted coffee increased with altitude in PN (2685.71, 2724.03, and 3847.14 µM trolox/g); meanwhile, the opposite was observed in N. High sensory scores were obtained in high altitudes. Alcohols and acids in roasted beans increase with altitude. *Leuconostoc* and *Pichia* showed potential as future coffee starters.

## 1. Introduction

Coffee quality involves various aspects, such as good practices and healthy raw fruits. For example, coffees with an SCA score of 80 or above are classified as high quality and speciality [1], making farmers aware of their practices during harvesting and processing. A well-taken care of post-harvesting results in high-quality coffee sold at higher prices. Though this is the primary key to high quality, coffee is a terroir product. Other factors such as genetic traits, geographic location, and climate affect microbiota, chemical compounds, sensory attributes, and quality [2,3,4].

Three processing methods are frequently performed worldwide: natural, pulped natural, and wet. The natural method uses whole intact mature fruits taken directly to dry in suspended terraces or concrete patios. Instead of whole intact fruits, the pulped natural method uses what is left after removing their skin and pulp with machines [5]. Similarly, the wet method uses the resulting depulped beans fermented underwater in tanks [6]. Drying is performed under the sun or in dryers until reaching 11 to 12% moisture, and afterwards, they are stored.

Continuously within the process, while coffee is drying, fermentation occurs. Fermentation generates essential flavor compounds and boosts quality and sensory scores. The natural microbiota consists of bacteria, yeast, and filamentous fungi. Some mechanisms performed by these groups during fermentation involve the degradation of pulp and mucilage by enzymes, pH lowering through organic acids production, filamentous fungi inhibition, and production of positive volatile esters and alcohols [5,7]. However, many other mechanisms and interactions are still unknown, yet the microbial population varies with the processing method.

Although all coffees are fermented, not many farmers can control this phase. As a result, new fermentation methods were innovated over the years. The most recent one that has contributed positively is known as self-induced anaerobic fermentation (SIAF), studied by Da Mota et al. [8], Martins et al. [9], and Pereira et al. [10]. This method offers an anaerobic-controlled environment through the natural production of carbon dioxide by microbial metabolism within closed bioreactors and is a water-free friendly method that produces high-quality coffees.

Upon the techniques to discriminate the microbial communities in fermented foods, a non-culture dependent is the most efficient since they are based on direct DNA detection. The polymerase chain reaction–denaturing gradient gel electrophoresis (PCR-DGGE) allows microbial monitoring over time and detecting species not isolated from conventional techniques. Previous studies have implemented this technique to discriminate the bacterial and fungal ecology involved in the different processing methods of *Coffea arabica* [6,11,12,13] and *Coffea canephora* [12].

Molecular approaches in combination with bioactive, organic, and volatile compounds analysis are significant coffee quality determiners. Chromatographic techniques such as high-performance liquid chromatography (HPLC) and gas chromatography-mass spectrometry (GC-MS) identify and separate low quantities of volatiles and non-volatile compounds due to their sensibility. The Fourier transform infrared (FTIR) spectroscopy technique is fast, reliable, and easily performed, needs no pre-treatment, and evaluates biochemical fingerprinting and molecular structures [14,15]. Together these techniques with multivariate statistical analysis facilitate real-time measurement of critical quality attributes [16].

Therefore, to evaluate coffees from altitudes 800, 1200, and 1400 m fermented using SIAF and processed by the natural and pulped natural method, molecular, chemical, and sensory analyses and correlations between the microbiota and bio compounds were performed.

## 2. Materials and Methods

### 2.1. Coffee Fermentation

Ripe fruits of *Coffea arabica* cv Catuaí Vermelho IAC 44 were manually collected from different altitudes: 800, 1200, and 1400 m, in the Caparaó region in Brazil. Before fermentation, half of the collected fruits from each altitude were processed via natural (N) and the other half via pulped natural (PN), which consisted of depulping the fruits mechanically. Then, the fruits and beans (the depulped fruits) were transferred into 20 L bioreactors with lids and closed to perform self-anaerobic fermentation (SIAF). Fermentations were performed in triplicate for each altitude in each process. All fermentative processes were carried out simultaneously at a farm located at 1200 m. Then after 72 h of fermentation, they were placed in suspended terraces to dry and reach 12% moisture. Drying lasted 17 days for PN and 28 days for N coffees. Sub-samples of approximately 100 g were taken in duplicate from each altitude before fermentation (T0), 48 h of fermentation (T48), end of drying (TF- coffee with 12% moisture), and after roasting (R) for further analysis.

### 2.2. Microbial Profiling

#### 2.2.1. Total DNA Extraction and PCR Analysis

T0, T48, and TF coffee samples from all altitudes were used for DGGE analysis. Three grams of sample were vortexed with 20 mL Milli-Q water for 10 min, then the suspension was separated from the whole fruits and beans, transferred to a new tube, and centrifuged at 12,745 RCF for 10 min at 4 °C. The supernatant was discarded, and the remaining pellet was used for DNA extraction. Total DNA was extracted using the DNA Purification from Tissues protocol from QIAamp DNA Mini Kit (Qiagen, Hilden, Germany), following the manufacturer’s instructions. Later, the DNA quality was verified with a 0.8% agarose gel, and samples were stored at −20 °C.

The bacterial community DNA was amplified with primers 338fgc (5′-CGC CCG CCG CGC GCG GCG GGC GGG GCG GGG GCA CGG GGG GAC TCC TAC GGG AGG CAG CAG-3′) and 518r (5′-ATT ACC GCG GCT GCT GG-3′), which target the V3 region of the 16S rRNA gene. The fungal community DNA was amplified with primers NL1GC (5′ CGC CCG CCG CGC GCG GCG GGC GGG GCG GGG GCA TAT CAA TAA GCG GAG GAA AAG-3′) and LS2 (5′-ATT CCC AAA CAA CTC GAC TC-3′) which target a fragment of the D1-region 26S rRNA gene. The primers used were the same as in Evangelista et al. [5]. The PCR was performed in a final reaction volume of 25 μL containing 12.5 µL TopTaq buffer from the TopTaq master mix kit (Qiagen), 0.1 µM of each primer for bacteria, and 0.3 µM for fungi, and 5 μL of extracted DNA. The amplification was performed according to Ramos et al. [17] for bacteria and fungi. Then, the amplification products were analyzed by electrophoresis on 0.8% agarose gels.

#### 2.2.2. Denaturing Gradient Gel Electrophoresis (DGGE) Analysis

The PCR products (10 µL per sample) were separated in polyacrylamide gels [8% (*w*/*v*) acrylamide: bisacrylamide (37.5:1)] in a 1× TAE buffer with a DCode system apparatus (BioRad Universal Dcode Mutation Detection System, Richmond, CA, USA). Band separation was achieved with a 15–55% urea-formamide denaturing gradient for bacterial communities and 20–60% for fungal communities [5]. 100% denaturant corresponds to 7 M urea and 40% (*v*/*v*) formamide. Electrophoresis was conducted at 170 V/60 °C for 4 h (bacteria and fungi). The gels were stained with SYBR-Green (1:10.000 *v*/*v*) following electrophoresis for 30 min. The images were visualized and photographed using a Transilluminator.

Selected bands from the PCR-DGGE gels were excised with a sterile blade and placed in 50 μL of sterile Milli-Q water at 4 °C overnight to allow the DNA to diffuse out of the polyacrylamide matrix and re-amplified with the 338fgc and 518r primers for bacteria and NL1 and LS2 for fungi. The PCR products were sent for sequencing, and later those sequences were compared with those available in the GenBank-NCBI database through blasting.

### 2.3. Biocompounds Evaluation

#### 2.3.1. HPLC: Organic Acids

Whole and depulped coffee fruits organic acids from T0 and TF were evaluated. Three grams of samples were vortexed in Falcon tubes containing 20 mL 16 mM perchloric acid in Milli-Q water and left at room temperature for 10 min. The resulting suspension was separated from the whole fruits and beans, transferred to another tube, and centrifuged at 12,745 RCF for 10 min at 4 °C to obtain the supernatant. The supernatant pH value was adjusted to 2.11 using perchloric acid in a new tube and recentrifuged under the conditions above. Later the obtained supernatant was filtered through a 0.22 mm cellulose acetate membrane (Merck Millipore, Darmstadt, Germany) and directly injected (20 mL) into the chromatographic column.

An HPLC system (Shimadzu Corp., Kyoto, Japan) equipped with a detection system consisting of a UV–Vis detector (SPD 10Ai) and a Shimpack SCR-101H (7.9 mm 30 cm) column operating at 50 °C (temperature used to separate water-soluble acids) at a flow rate of 0.6 mL/min was used for analysis. The acids were identified by comparison with the retention times of authentic standards. The quantification was performed using calibration curves constructed with standard compounds. Malic and citric acid were purchased from Merck (Darmstadt, Germany). Lactic, tartaric, acetic, and succinic acids were purchased from Sigma-Aldrich (Saint Louis, MO, USA). All analyses were performed in duplicate.

#### 2.3.2. HPLC: Bioactive Compounds Trigonelline, Caffeine, and 5-Chlorogenic Acid (5-CGA)

Caffeine, 5-CGA, and trigonelline from T0, TF, and R were identified using a Shimadzu liquid chromatography system (Shimadzu Corp., Kyoto, Japan) equipped with a C18 column following the protocol in Bressani et al. [18]. Ground whole fruits and beans (0.5 g) were placed in tubes containing 50 mL Milli-Q water and boiled for 3 min to extract the total compounds. Then, the resulting suspension was filtered through a 0.22 µm cellulose acetate membrane (Merck Millipore). Identification and quantitative analysis were performed by injecting caffeine, trigonelline, and 5-CGA (Sigma-Aldrich) standards and building calibration curves. All analyses were performed in duplicate.

#### 2.3.3. Total Polyphenols and Antioxidant Capacity

Coffee samples from N and PN (T0 and R) were defatted following the methodology described by Batista et al. [19]. One hundred fruits and beans were ground with liquid nitrogen per sample, then 4 g were weighed. 20 mL of *n*-hexane (Merck) was added into the 4 g to separate the lipids from the supernatant three times. The resulting lipid-free samples were air-dried (25 °C) for 24 h to evaporate the residual organic solvent. Afterward, the polyphenols and antioxidants were extracted according to Kim et al. [20], with slight modifications. Fifty milliliters of distilled water at 90 °C were added to a tube containing 2.75 g lipid-free ground coffee. Then, the mixture was left at room temperature (25 °C) for 20 min. After that period, the mixture was filtered through a Whatman No. 2 filter paper.

The total polyphenols content (TPC) was determined through a spectrophotometric assay (UV-VIS Spectrum SP-2000 UV, Biosystems, Singapore) following the Follin—Ciocalteau methodology [21]. Coffee extract (500 mL), 2.5 mL of Folin–Ciocalteau reagent (10%), and 2.0 mL of Na_2_CO_3_ (4% *w*/*v*) were homogenized and incubated at room temperature in the dark for 120 min. The absorbance of the samples was measured at 750 nm. The TPC concentrations were calculated based on the standard curve of gallic acid (ranging from 10 to 100 µg/mL) and expressed as milligrams of gallic acid equivalents per gram of ground coffee (mg GAE/g). All analyses were performed in triplicate.

Two different assays were performed to measure the coffee extract’s antioxidant capacity. The first assay, known as the 2,2-diphenyl-1-picryl-hydrazyl-hydrate (DPPH) radical scavenging assay, was performed as follows: 0.1 mL of coffee extract was added to 3.9 mL of the DPPH radical solution (0.06 mM) and incubated at room temperature, in the dark for 120 min, then the absorbance was measured at 515 nm (Spectrophotometer UV-Vis Spectrum R SP-2000UV, Shanghai, China). Trolox was used as a standard. A calibration curve (y = −0.0004x + 0.6636) was assembled using a range of 10, 20, 30, 40, 50, and 60 mM Trolox with linearity R^2^ = 0.9999. The results were expressed as µM × Trolox Equivalents (TE) per gram of ground coffee (µM × TE/g). The second assay, known as 2,′-azinobis (3-ethylbenzothiazoline-6-sulfonic acid) (ABTS), was performed with ABTS stock solution reaction (7 mM) and potassium persulfate (140 mM). Later, the mixture was left in the dark at room temperature for 16 h before use. The ABTS solution was diluted in ethanol to an absorbance of 0.70 ± 0.05 at 734 nm. Thirty microliters of the coffee extracts were added to 3.0 mL of the ABTS radical solution, and after 6 min, the absorbance was measured. Trolox was used as a standard. A calibration curve (y = −0.0003x + 0.6802) was assembled using a range of 100, 500, 1000, 1500, and 2000 mM Trolox with the linearity of R^2^ = 0.9983. The results were expressed as mM × Trolox Equivalents (TE) per gram of ground coffee (µM × TE/g).

#### 2.3.4. GC-MS: Volatiles

Volatile compounds were extracted from T0, TF, and R samples using headspace-solid phase microextraction (HS-SPME). The compounds were analyzed using a Shimadzu QP2010 GC model equipped with mass spectrometry (MS) and a silica capillary Carbo-Wax 20 M (30 m × 0.25 mm × 0.25 mm) column. The operating conditions were performed as described by Martinez et al. [3]. The volatile compounds were identified by comparing the mass spectra to the NIST11 library. In addition, an alkane series (C7–C40) was used to calculate each compound’s retention index (RI) and compare it with the RI values found in the literature. Lower alkanes were determined via an automatic integration performed by the program used for peak analysis.

#### 2.3.5. Profiling by FTIR

According to Liang et al. [16], FTIR analysis was conducted with minor modifications. Ground coffee samples (2 g) from T0, T48, TF, and R were frozen for 24 h at −20 °C. Then, the samples were lyophilized for 24 h. Coffees’ FTIR spectra were recorded on a Digilab Excalibur, series FTS 3000 (Digilab, Randolph, MA, USA), coupled to an attenuated total reflectance (ATR) accessory equipped with a ZnSe reflection crystal. The spectra were acquired at room temperature with 32 scans/samples in the range of 4000 to 400 cm^−1^ at a resolution of 4 cm^−1^.

### 2.4. Sensory Perception: Cup Test and Attributes

Coffee roasting from each altitude was performed within 8:30–9:30 min until obtaining beans with a coloration between #55–#65 on the Agtron scale. The roasted beans were ground in an electric mill (Mahlkönig, EK43 model, Hamburg, Germany) to a slightly thicker size than the one used for drip brew (70–75% of particles should be able to pass through a US Standard size 20 mesh sieve). With a Q-Grader Coffee Certificate, a panel of five expert coffee tasters performed sensory analysis using five cups for each sample. The attributes evaluated were fragrance/aroma, flavor, aftertaste, acidity, body, balance, uniformity, sweetness, clean cup, overall, and the total score apart from the sensory descriptors.

### 2.5. Statistical Analysis

Organic acids were analyzed with the Tukey test at 5% significance using the SISVAR software [22]. Trigonelline, caffeine, 5-CGA, total phenolics, and DPPH and ABTS assays were analyzed with the Tukey test at 5% significance using the SISVAR software. Volatile principal component analysis (PCA) was performed using the XLSTAT 2021 software for each altitude and process. Pearson correlations and PCAs between DGGE identifications and chemical compounds were performed using a presence-absence matrix for DGGE identifications (T0, T48, and R) and total acids concentrations and relative volatiles concentrations for each altitude and process with XLSTAT 2021 and Origin 2021b software.

## 3. Results

### 3.1. Microbial Communities

Thirteen (13) microorganisms were identified from the bacterial PCR-DGGE (Figure 1). The bands were more intense at T48 than at T0 and TF. The number of bacteria identified lowered with an increase in altitude in the N process (800 (10), 1200 (8), and 1400 m (6)). Meantime the number of bacteria identified in the PN process increased with altitude (800 (9), 1200 (13), and 1400 m (12)). Overall, more bacteria in the PN process remained from the beginning to the end of drying. At 800 m, bacteria #2 (uncultured *Leuconostoc* sp.) and 8 (*Pseudomonas* sp.) were present from T0 to TF in both processes. Other microorganisms from 800 m always present include bacteria #1 (*Weissella confusa*) and 6 (uncultured *Serratia* sp.) in N, and #4 (uncultured *Streptomyces* sp.), 5 (uncultured bacterium), and 13 (uncultured *Serratia* sp.) in PN. Following, the only bacteria common in both processes at 1200 m present from T0 to TF was #3 (uncultured *Leuconostoc* sp). Similarly, at 1400 m, bacteria #3 was also in both processes, with #8 and 13 remaining, along with fermentation and drying. Bacteria #12, belonging to an uncultured *Pantoea* sp. (Figure 1), was only present at 1200 PN T0. In the PN process, bands of bacteria #1 appeared from 48 h and stayed until TF for all altitudes.

More microorganisms were identified in the fungal DGGE (19) (Figure 1). All altitudes and processes shared a similar profile of bands, especially at T48 bands were more intense (Figure 1). The number of fungi identified increased with altitude in the N process (800 (11), 1200 (15), and 1400 m (15)). Meanwhile, it was the opposite in PN. When the processes were compared, no fungi took part from T0 until the end in altitudes 800 and 1200 m, and only at altitude 1400 m had fungus #13 (*Cladosporium flabelliforme*) in common. The fungi that remained from T0 to TF in the N process at 800 and 1200 m were #1 (*Hanseniaspora uvarum*), 5 (*Wickerhamomyces anomalus*), 12 (uncultured fungus), 16 (*Pichia terrícola*), 17 (*Pichia fermentans*), and 19 (*Pichia kudriavzevii*) and #5, 10 (*Cladosporium delicatulum*), and 12 respectively. Moreover, the fungus present from T0 to TF in N at 1400 was *Aureobasidium* sp. (#3). In the PN process, fungi #18 (uncultured *Pichia*) and 19 only appeared at 48 h for all altitudes.

### 3.2. Organic Acids

The detected acids (acetic, malic, citric, lactic, succinic, and tartaric) are shown in Figure 2A. Within the N process, there were no statistical differences between 800 and 1200 m, only in 1400 m (Figure 2A). However, there was no statistical difference between the altitudes (Figure 2A) for the PN method. However, when each process was compared for each altitude, only 800 and 1200 m showed statistical differences.

Moreover, Figure 2B shows the *p*-value differences obtained from the interactions of the acids and the evaluated times, altitudes, and processes. Finally, T0 and TF presented statistical differences among the acids. Similarly, only altitudes 800 and 1200 presented statistical differences and the N process when the processes were compared.

Independent of the time, acetic acid was found in higher concentrations in N than PN, mainly at N 800 m (T0: 14.43 and TF: 13.47 g/kg) (Figure 2A). The acetic acid decreased from T0 to TF in all N altitudes except for 1200 m (T0: 8.07 and TF: 10.09 g/kg). In PN, acetic acid was only detected at T0, presenting the highest concentration at 800 (6.48 g/kg). After acetic, malic acid was the second most predominant acid, followed by citric acid. In both processes, malic and citric acid decrease from T0 to TF. Lactic was only detected at TF for all altitudes in N (800, 1200, 1400 m: 7.14, 3.06, 1.07 g/kg) and PN (800, 1200, 1400 m: 2.98, 2.81, 1.10 g/kg). Tartaric acid was detected in all altitudes in both processes except at N 800. Succinic acid increased along the processing time in N at 800 (T0: 1.21–TF: 1.67 g/kg) and PN at 1200 m (T0: 1.04–TF: 1.70 g/kg), while at the other altitudes, it decreased.

### 3.3. Bioactive Compounds, Total Polyphenols, and Antioxidant Capacity

Overall, only trigonelline showed a statistical difference within each process due to the low concentrations compared with the other bioactive compounds, as observed in Figure 2C,D. In the N process (Figure 2C), caffeine concentrations increased after roasting for all altitudes (800: 16.57, 1200: 14.81, and 1400: 16.45 g/kg). Also, trigonelline at 800 (8.34 g/kg), 1200 (9.86 g/kg), and 1400 (9.00 g/kg), and 5-CGA at 800 (15.54 g/kg) and 1400 (18.45 g/kg). Within the PN process (Figure 2D), caffeine also increased after roasting in altitudes 800 (10.19 g/kg), 1200 (14.24 g/kg), and 1400 (10.84 g/kg). Moreover, 5-CGA only increased after roasting at 800 m (11.98 g/kg), while trigonelline increased at 800 (7.65 g/kg) and 1200 m (4.19 g/kg).

When the processes were compared, only 800 m showed a statistical difference (Figure 2C,D) because 5-CGA at T0 was the only sample that presented differences (N: 14.86 and PN: 1.40 g/kg). When the bioactive compounds were compared within each process, altitude, and time, only TF in N 1200, TF and R in PN 1200, and TF in PN 1400 showed differences. That is, trigonelline (7.00 g/kg), caffeine (9.92 g/kg), and 5-CGA (19.62 g/kg) concentrations in N 1200 TF were all statistically different (Figure 2C). In PN 1200 TF, trigonelline and caffeine concentrations differed similarly from 5-CGA (21.15 g/kg) (Figure 2D). All bioactive compound concentrations in PN 1200 R were statistically different. In sample PN 1400 TF, 5-CGA (26.49 g/kg) was the compound with the highest concentration.

There were no statistical differences between the two processes regarding total polyphenols and antioxidant capacity. Compared to T0, after roasting, the total polyphenols concentrations and antioxidant capacity increased for all altitudes in each process (Figure 2E,F). ABTS capacity was higher after roasting for all samples than DPPH capacity. There was a statistical difference between T0 and R for all altitudes in each process, as observed in Figure 2E,F. When T0 of all altitudes was compared in each process, no statistical difference was observed in total polyphenols and antioxidant capacity. When concentrations at R were compared, only an altitude of 1400 for each process was different (N: total polyphenols-704.74 mg/g, DPPH-2494.71 µM·TE/g, and ABTS-2775.76 µM·TE/g and PN: total polyphenols- 735.44 mg/g, DPPH-2881.49 µM·TE/g, and ABTS-3847.14 µM·TE/g).

### 3.4. Volatiles

One hundred and twenty-one (121) volatile compounds were identified and classified into 16 chemical groups (acids, alcohols, aldehydes, alkanes, anhydrides, esters, free fatty acids (FFA), furans hydrocarbons, ketones, lactones, phenols, pyrans, pyridines, pyrroles, and thiophenes) (the list is displayed on Appendix A). Among the total compounds, 8 (S)-3-ethyl-4-methylpentanol, 1-heptanol, 1-hexanol, 2,3-butanediol, 2,4-decadienal, 2-phenyl-2-butenal, 1,2-cyclopentanedione, 3-methyl-, and acetoin) were only produced by yeasts. Figure 3A shows the volatiles of PCAs at T0, TF, and R of all altitudes in N and PN. The volatiles were affected mainly by time and not by altitude. Among the volatiles showing the highest relative concentrations in the N process, 7 were correlated with T0 (1Al, 2Al, 7Al, 10Al, 13Al, 15Al, and 10Ad), 3 with TF (12Al, 20Al, and 6P), and 5 with R (1Ac, 6Ad, 12Ad, 2F, and 7K) as observed in Figure 3A. From the volatiles that had the highest relative concentrations in the PN process, 7 correlated with T0 (1Al, 2Al, 7Al, 10Al, 13Al, 15Al, and 10Ad), 9 with TF (12Al, 20Al, and 6P), and 5 with R (1Ac, 12Ad, 6Ad, 2F, and 7K).

The chemical groups that dominated the coffees at T0 and TF were alcohols, aldehydes, acids, esters, and phenols, and at R were furans, aldehydes, acids, ketones, and esters, as observed in Figure 3B. The alcohol’s total relative concentration percentage increased with altitude within each process at T0 and was observed higher percentages in N than PN (N: 800: 71.1%, 1200: 80%, and 1400: 82.5%, PN: 800: 48.2%, 1200: 65.7%, and 1400: 73.1%). Similar behavior was observed for the PN TF (800: 52.9%, 1200: 55%, 1400: 58.6%), and at 1400 N, TF presented the highest percentage (56.8%). The aldehydes group in T0 was higher than TF and increased with roasting in N and PN. Acids increased with roasting for both processes and within R with altitude (N: 800: 13.4%, 1200: 18.3%, 1400: 18.7%, PN: 800: 14.3%, 1200: 17.8%, and 1400: 18.6%). Esters were in higher percentages in PN than N process. Phenols in TF were higher than at T0 and R, mainly in the N process (800: 16.6%, 1200: 18.1%, and 1400: 12.1%). Furans were only present in 800 PN T0 (0.1%) and all roasted samples in both processes. While altitude increases, the total relative concentrations decrease (N: 800: 36.7%, 1200: 35.2%, 1400: 31.1%, PN: 800: 35.2%, 1200: 32.7%, and 1400: 32.7%). Ketones also increased with roasting, mainly in the N process (800: 6.4%, 1200: 8.1%, and 1400: 6.3%).

Other groups in Figure 3B include FFA, which were found in higher percentages at TF than T0 and R, mainly in the PN process (%: 800: 6.7, 1200: 9.5, and 1400: 6.1). Pyridines and thiophenes were only present in roasted samples. Pyrroles were only present in N and PN roasted samples and at TF in the N process (%: 800: 0.1, 1200: 0.1, and 1400: 0.1).

### 3.5. Correlation between the Microbial Communities and Chemical Compounds (Organic Acids and Volatiles)

The correlation and grouping of DGGE bands and compound concentrations are presented in Figure 4A,B. When the bacteria and organic acids were correlated (Figure 4A), a highly positive correlation was obtained between *W. confusa* and acetic (0.82) and lactic acid (0.91). Bacteria #6 was positive (0.83), and #3 (−0.71) negatively correlated with lactic acid. Bacteria #7 (uncultured bacterium) and 9 (uncultured *Klebsiella* sp.) were positively correlated with tartaric acid. Also, bacteria #3 showed a strong negative correlation with lactic acid (−0.71). Malic (−0.87) and succinic acid (−0.94) presented strong negative correlations with *Pseudomonas* sp. When the fungi and organic acids were correlated (Figure 4A), *H. uvarum* (0.81), *W. anomalus* (0.92), an uncultured fungus (0.89), and 4 genera of *Pichia* were strongly correlated with acetic acid (*P. terricola* 0.81, *P. fermentans* 0.81, *P. kudriavzevii* 0.75, and *P. kluyveri* −0.91). Additionally, *H. uvarum* (0.91), *P. terricola* (0.86), and *P. fermentans* (0.86) were strongly correlated with lactic acid together with *Candida* sp. (−0.91). *M. caribbica* was positively correlated with succinic (0.94), malic (0.87), and citric acid (0.93). *S. cerevisiae* was strongly correlated with malic (0.90) and citric acid (0.81).

Most volatile groups depend more on fungi than bacteria (Figure 4B). The chemical groups alcohols and acids were not clustered with any bacteria. Instead, they were clustered with more than half of the identified fungi (10). Three groups resulted from the PCA of bacteria and volatiles groups, as seen in Figure 4B. Bacteria #1, 6, and 2 were grouped with furans and ketones. Esters, aldehydes, FFA, pyrans, hydrocarbons, anhydrides, and thiophenes were clustered with the rest of the bacteria except for bacteria #13 and 3. When the fungal PCA was analyzed with the volatile chemical groups, *P. terricola*, *P. fermentans*, *P. kudriavzevii*, *H. uvarum*, *W. anomalus*, and an uncultured fungus were related to groups ketones, furans, lactones, alkanes, phenols, and pyrroles. The uncultured *Pichia* was grouped with esters and thiophenes, and *P. kluyveri* and *C. flabelliforme* with aldehydes, hydrocarbons, FFA, and anhydrides.

### 3.6. FTIR Profile

FTIR’s spectral infrared analysis from certain functional groups of green and roasted coffee is shown in Figure 5. The ATR-FTIR spectra revealed that the intensity of some bands changes during fermentation and roasting. The band at wavenumber 809 cm^−1^ was derived from cyclohexane C−O twisting, and 1120 and 1165 cm^−1^ were derived from cyclohexane CH, C−OH bending, and the phenyl ring bending vibration, respectively [16]. According to Liang et al. [16], all those three bands plus band 1032 cm^−1^ are the characteristic bands of pure chlorogenic acid isomers. In the N process (Figure 5A–C), the intensity of band 809 cm^−1^ increases with roasting except at altitude 800. Similar results were obtained for the band 1032 cm^−1^. On the contrary, the other bands’ intensity increased after roasting for all altitudes. In the PN process (Figure 5D–F), the intensity of the mentioned bands increased after roasting.

Lipids exhibit a characteristic band from the carbonyl C=O vibration at ~1744 cm^−1^ in arabica [23]. Moreover, that band and the band at ~1150 cm^−1^ change remarkably in arabica coffee. Both bands’ intensity increased after roasting in all altitudes in the N process, being more intense at 1400. The same result was observed in the PN process.

Based on Bressani et al. [24], the band at 1457 cm^−1^ is associated with C–H bending vibration belonging to CH_2_ and CH_3_ aliphatic groups. This band increases its intensity after roasting for all altitudes in both processes. The band at 1603 cm^−1^ is associated with C=O vibrations, characteristic of aromatic compounds [25]. In both processes, this band was intense at T0, except in 800 N, which was intense at T48.

The 2923–2852 cm^−1^ region is associated with symmetrical and asymmetrical vibrations of C-H groups of carbohydrates and caffeine. The intensity of this region increases after roasting the samples of N and PN, mainly at 1400. Another region to be considered is 3500–3200 cm^−1^ which corresponds to the stretching vibrations of O-H or N-H functional groups, carboxylic acids, amines, amides, and alcohols [26]. In the N process, the intensity of this region was different for each altitude. At altitude 800, the intensity was high at T48 (Figure 5A), 1200 T0 (Figure 5B), and 1400 R (Figure 5C). In the PN process, the intensity was high at R in altitudes 800 and 1400, and at altitude 1200, it was high at T0.

### 3.7. Sensory Response

The highest scores obtained from the cup test were for the N coffees compared to the pulped natural coffees, mainly at 1400 m, with 88.13 (Figure 6). Coffee N 1400 and PN 800 had a richer descriptors profile than the other altitudes in each process. Raspberry, cocoa, rum, lychee, pear, roses, and hops were among the descriptors only found at N 1400.

The frequency in which the attributes were perceived is also shown in Figure 6. Among the primary descriptors, the most frequently perceived were chocolate, molasses, caramel, brown sugar, and fruity. Floral and spices were perceived in all samples, and floral was the highest in PN 1200 (5%) and spices in N 800 (8%). The attribute cereal was only perceived in PN 800 (2%). Only sample PN 800 had most of the descriptors from the overall category, including medium/slightly rough (2%), slightly rough (2%), dry (2%), rough (2%), and short (3.9%). The other samples presented the attribute pleasant/long or pleasant. Between the acids, citric acid was present in all samples and had a high frequency (8%) in sample N 1200. A creamy body was only perceived in PN 800 (2%), while samples PN 1200 (2.5%) and PN 1400 (1.9%) presented a silky body, and only N 1200 presented a velvety body (2%).

The sample with the highest sensory score (N 1400) had exotic and diverse attributes such as cashew, lychee, rum, vanilla, jasmine, pear, roses, raspberry, and hop. Only samples N 1200 ginger and N 800 dulce de leche were perceived.

## 4. Discussion

The intensity of bands from PCR-DGGE at T48 was expected since it is a time when the coffee microbiota population is high and dominates, as seen in Silva et al. [27] and Vilela et al. [13]. The lactic acid bacteria (LAB) group comprehends many genera, such as *Leuconostoc*, *Fructobacillus*, *Weissella*, *Lactobacillus*, *Pediococcus*, *Lactococcus*, and *Enterococcus*. In coffee, the LAB group metabolizes sugars and other minor compounds into lactic acid and some into acetic acid, which decreases and assists by breaking down pectin. Out of the 13 detected bacteria, 3 (*Weissella confusa* and two *Leuconostoc* sp.) belonged to the LAB group, increased with 48 h, and remained until coffee dried, demonstrating its dominance in the N and PN process. Another species of *Leuconostoc* have been frequently identified in the PN and N, which is *Leuconostoc mesenteroides* [5,13], showing 100% abundance among the isolated colonies from coffees at 800, 1200, and 1400 m altitude in Martins et al. [9]. The intense bands of all detected LAB until TF may explain the increase of lactic acid and confirm the strong correlation between *W. confusa* and uncultured *Leuconostoc* sp.

Four of the bacteria identified belonged to the Enterobacteriaceae family (*Serratia* sp., *Klebsiella* sp., and *Enterobacter* sp.), known as pathogens, the most derived from human contact. The genus *Klebsiella* and *Serratia* has been previously identified by Silva et al. [27] and Van Pee et al. [28] using the natural method and by Vilela et al. [13] using the pulped natural method as in this study. Many functions of *Pseudomonas* sp. in coffee are unknown, except that it is a caffeine-degrading bacteria through an N-demethylation reaction [29]. Although its correlation with caffeine was not evaluated, we did not observe a decrease after roasting in N and PN, only from T0 to TF in N 800 and N 1400, where *Pseudomonas* sp. bands were also very intense. In addition, the intense bands of this genus may be related to the high malic acid concentration in T0 and TF due to their strong correlation.

Yeast growth is achieved by consuming essential compounds such as fermentable sugars, amino acids, vitamins, and minerals [30]. In coffee, yeast improves the quality of low-altitude coffees [24] and produces organic acids such as citric and succinic acid since they derive from the Krebs cycle [31]. According to Martinez et al. [32], *S. cerevisiae* in coffee can increase the citric acid content.

Apart from producing organic acids, yeasts are mostly known for producing volatile compounds. *Non-Saccharomyces* yeasts possess characteristics that in *S. cerevisiae* are absent, such as high aromatic compounds such as esters, higher alcohols, and fatty acids [30]. This statement was confirmed with the clustering in the PCA of some yeast from the genus *Pichia* with the esters and FFA group. Other *Pichia* strains in this study affected the furans group. The *Pichia* genus dominates cocoa fermentation and is a prominent starter culture, and it is a widespread presence of this genus in different coffee processes. Species such as *P. kudriavzevii* have only been detected in wet process coffees [33,34], and *P. kluyveri* has been detected in pulped natural coffees fermented by the SIAF method [9,10]. *H. uvarum* is considered fermentative, dominating coffee processes in Brazil, Ecuador, Australia, Cameroon, and Tanzania [9,12,13,33,34,35,36]. *H. uvarum* was present at N and PN 800 coffees in Martins et al. [9] but not in coffees from 1200 and 1400. In contrast, *H. uvarum* was detected in this study in all altitudes and processes until T48, lasting until TF only in N 800. Additionally, although the filamentous fungi detected are aerobic, in this anaerobic process, they do not grow, but their spores may remain.

All acids concentrations were higher in the N coffees for all altitudes, which was expected since the process uses whole intact fruits. Therefore, the composition of the total compound is maintained and not diluted as in the PN coffees. Lactic, acetic, and succinic acid concentrations are affected by the altitude in the N process since they decrease with an increase in altitude. Though no acetic acid bacteria were identified, acetic acid was the most abundant and probably produced by other microorganisms, as stated above. Like this study, acetic acid was dominant in semi-anaerobic fermented N coffees and spontaneous anaerobic fermented 600 m N coffees [18,24]. Citric and malic acid is naturally found in coffee. They are wanted because they confer tartness, fruity, and berry flavors. Both acids were in higher concentrations than what is usually detected in coffees processed via PN, as observed in Evangelista et al. [5]. Malic acid decrease in both processes may have occurred through converting malic acid to lactic acid by naturally occurring bacteria in coffee [8,37]. Since lactic acid was only detected at TF in all treatments is considered a microbial-derived product in coffee fermentation, and its permanence after drying indicates that the responsible microorganisms are still active.

Altitude modifies the concentration of bioactive compounds by affecting the physiology and the microbial communities [3,38]. At high altitudes, temperatures are lower, resulting in lower metabolisms and extended maturation [38]. Caffeine is an alkaloid nitrogenous secondary metabolite that contains bitterness in the beverage, is formed in immature coffee fruits, and gradually accumulates during seed development [39]. According to Worku et al. [38], lower-altitude coffee beans contain more caffeine and GCA. based on the obtained results, after roasting, the lower-altitude coffee (from 800 m) presented the highest caffeine concentration only in the N process. In Bressani et al. [18], caffeine concentrations were much lower than those found in the roasted beans from the N process in this study, probably due to the composition of the raw fruits, coffee processing, and fermentation method.

CGA is the most abundant phenolic compound in green beans and belongs to a family of esters. During roasting, a large percentage of CGAs degrade into caffeic acid, lactones, and phenols derivates, and their content is subjected to gene expression at different stages of beans and temperatures [39]. Opposite to the caffeine results, 5-CGA after roasting increased with altitude only in the N process. That might be why only 5-CGA showed a statistical difference. The FTIR analysis also showed that most CGA bands increase with roasting for the N and PN processes. Similar behavior happened for the bands characterizing caffeine. Trigonelline is a pyridine derivative known to contribute indirectly to the formation of appreciated flavor products, including furans, pyrazine, pyridines, and pyrroles, during coffee roasting. After roasting, the trigonelline concentration increased at all altitudes except at the highest altitude in PN. Bertrand et al. [40] reported that green coffee beans grown at high altitudes and processed via the wet method in Costa Rica have low trigonelline contents. Our study obtained the same results: that is, before green beans were fermented for both processes and for TF in N. Although there was no statistical difference, there was a trigonelline decrease with an altitude increase.

As expected, the alcohols group was the most abundant in green beans, be it unfermented (T0) or fermented (TF)—also, the group decreased after drying, which may be due to their volatilization. No direct studies explain the increase of this group with altitude. The FTIR bands corresponding to some alcohol vibrations in N coffee behaved differently by dominating a time at each altitude. Similar behavior was observed for PN 1200. Each time, 20Al was detected as phenylethyl alcohol within the characteristic compounds. Phenethyl alcohol is a higher alcohol that produces rose-like flavors and is generated during phenylalanine metabolism, mainly by yeasts [41]. Esters are found in low concentrations and influence coffee flavor by generating fruity flavors; they are mainly produced by acid catalysis and decreased pH and are derived from yeasts. The esters group was in high concentrations like alcohols, which was expected due to the versatility of attributes perceived during the tasting.

Pyrroles and furans are aromatic groups that contribute to floral, rose-like, smoky flavors and caramel-like odor. In this study, they were abundant in roasted coffee independent of the coffee process. Green arabica beans contain a crude lipid of 14–18% (*w*/*w*), and about 1–2% are lost during roasting. Certain FFA is probably lost when the skin and pulp are removed from the natural process, so it is perceived in higher percentages in PN.

Sensory evaluation is biased personal appraisal. Due to this, three or more tasters are mandatory, although six or more would be ideal for SCA protocol, according to Pereira et al. [42]. The richness of bacteria at 1400 could have influenced the diversity of attributes detected and scores. In addition, at higher altitudes, coffee metabolism is much slower, which conduces to a higher compound accumulation and consequently increases the beverage quality [2]. However, not many studies discriminate in detail the attributes perceived, and this is the first study carrying out this type of discrimination. Pleasant attributes that contribute to sweetness took part in the main descriptors. Having fruity and citric attributes in high frequencies is a positive indicator of coffee quality.

## 5. Conclusions

The LAB group dominated the SIAF fermentations at all altitudes. The SIAF fermentation from different coffee processes and altitudes impacted the acid profile. The concentrations of lactic and acetic acid present in this work were the right amount for obtaining specialty coffees. Altitude 1200 at the end of both processing methods affected the bioactive compounds, mainly 5-CGA. Independent of the altitude, different volatile groups dominated before fermentation, after drying, and roasting. The alcohols and acids group in roasted beans increase with altitude. Altitude 1200 is adequate for compound modulations by using the SIAF method. ABTS capacity in roasted coffee increased as altitude increased in PN (2685.71, 2724.03, and 3847.14 µM trolox/g); meanwhile, the opposite was observed in N. High-quality specialty coffees were produced with the SIAF method, obtaining higher sensory scores in high altitudes. Microorganisms from the genera *Leuconostoc* and *Pichia* showed potential as future coffee starters since they were correlated with the main volatile precursors of flavors and lactic and acetic acid.

## Figures and Tables

**Figure 1 foods-11-03945-f001:**
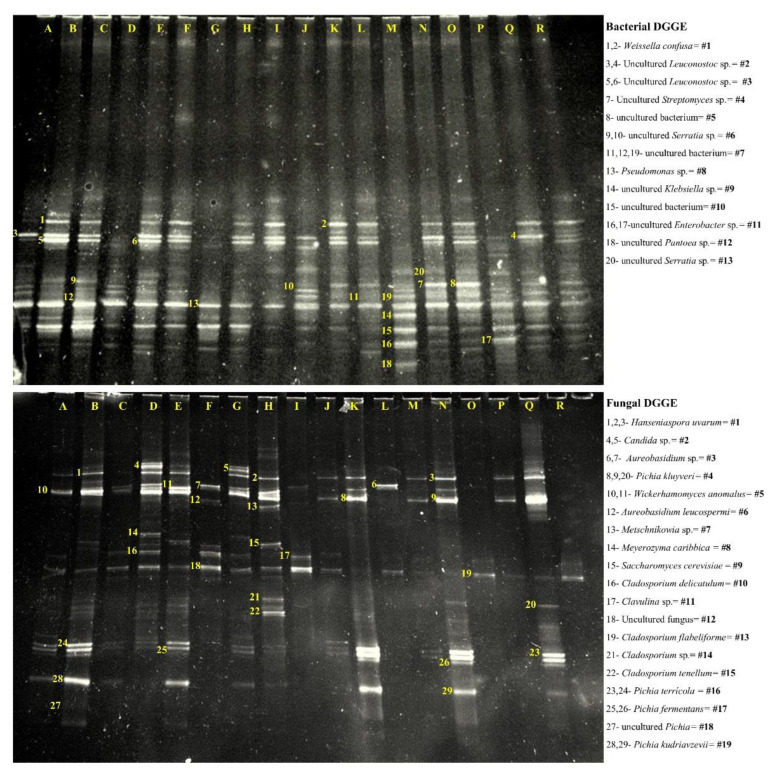
DGGE microbial profile of bacteria and fungi at different times and altitudes. The numbers over the bands correspond to the identified microorganism on the right. The upper image corresponds to the bacterial profile, and the lower image corresponds to the fungal profile. N process: 800 (A, B, C), 1200 (D, E, F), and 1400 m (G, H, I), and PN process: 800 (J, K, L), 1200 (M, N, O), and 1400 m (P, Q, R). T0 (A, D, G, J, M, and P), T48 (B, E, H, K, N, and Q), and TF (C, F, I, L, O, and R).

**Figure 2 foods-11-03945-f002:**
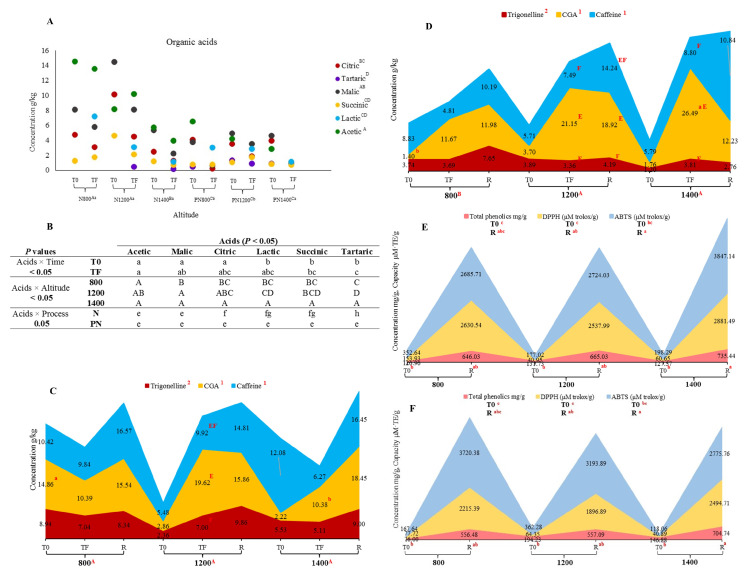
(**A**) Organic acid concentrations detected before fermentation and after drying from coffees proceeded via natural and pulped natural at different altitudes (800, 1200, and 1400 m). N = natural process, and PN = pulped natural process. T0 = before fermentation, and TF = after drying. Different uppercase letters in the altitudes represent statistical differences at *p* < 0.05 in each process: ^A, B^ = N, and ^C^ = PN. Different lowercase letters in the altitudes represent statistical differences at *p* < 0.05 between the processes for each altitude: ^a,b^. Different uppercase letters on the acids represent statistical differences between them at *p* < 0.05: ^A, AB, BC, CD, D, D^. Standard deviations: 800 (N: 0–5.67 and PN: 0–4.33), 1200 (N: 0–3.03 and PN: 0–3.83), and 1400 m (N: 0.06–1.38 and PN: 0–3.09). (**B**) Represent all the statistical differences obtained from the interactions with the acids when the Tuckey test at *p* < 0.05 was applied. In Acids × Time: T0 = before fermentation and TF = after drying, different lowercase letters were used to represent the differences between the acids each time. In Acids × Altitude: different uppercase letters were used to represent the differences between the acids at each altitude. In Acids × Process: N = natural process and PN: pulped natural process. Different lowercase letters were used to represent the differences between the acids in each coffee process. (**C**,**D**) Trigonelline, 5-CGA, and caffeine concentrations before fermentation, drying, and roasting. (**C**) N process and (**D**) PN process. Different numbers on the legend represent statistical differences at *p* < 0.05 obtained from the Tuckey test between the bioactive compounds. Different uppercase letters on the altitudes represent statistical differences at *p* < 0.05 when both processes in each altitude are compared. Different lowercase letters inside the graph represent statistical differences at *p* < 0.05 when the processes, altitudes, and times were compared. Upper-case letters inside the graph represented statistical differences at *p* < 0.05 when the compounds within each process, altitude, and time were compared. (**E**,**F**) Total phenolics concentration and DPPH and ABTS antioxidant capacity. (**E**) N process and (**F**) PN process. Different lowercase letters on the legend represent statistical differences at *p* < 0.05 obtained from the Tuckey test when compounds and times were compared. Different lowercase letters in the times below within the graph represent statistical differences at *p* < 0.05 when the times at each altitude were compared.

**Figure 3 foods-11-03945-f003:**
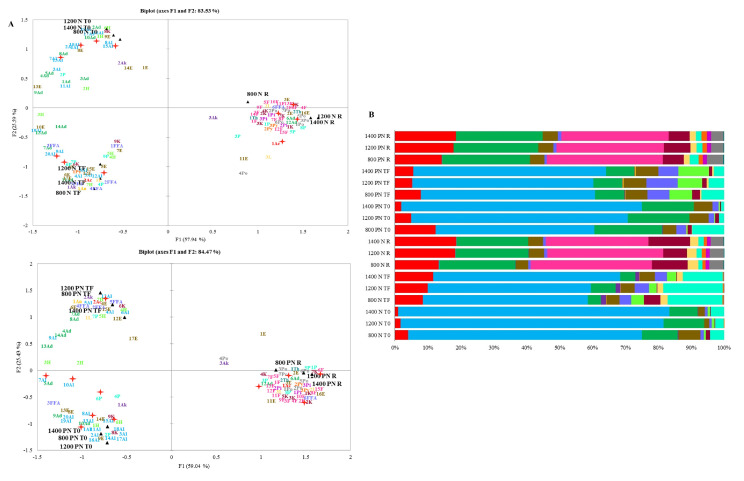
(**A**) Principal component analysis (PCA) of all the volatile compounds detected from each process. The upper graph represents the N process, and the low graph represents the PN process. 
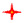
 Represents the volatiles with the highest relative concentrations and correlation with T0, TF, and R. T0: before fermentation, TF: after drying, and R: after roasting. The compound name for each code (1Ac - 2Th) is displayed on Appendix A. (**B**) Total relative concentration percentage of the chemical groups detected in the N and PN process and altitude. 
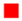
 acids, 
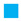
 alcohols, 

 aldehydes, 
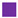
 alkanes, 
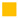
 anhydrides, 
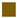
 esters, 
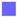
 FFA, 
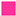
 furans, 
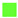
 hydrocarbons, 
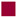
 ketones, 
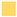
 lactones, 
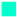
 phenols, 
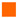
 pyrans, 
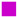
 pyridines, 
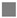
 pyrroles, and 
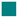
 thiophenes.

**Figure 4 foods-11-03945-f004:**
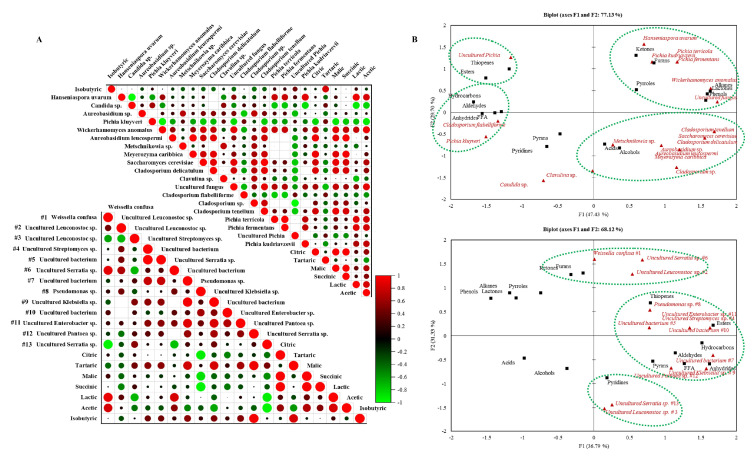
(**A**) Down-stream Pearson correlation matrix of bacteria and organic acids. Down-stream Pearson correlation matrix of fungi and organic acids. (**B**) Principal component analysis (PCA) of all the detected volatile groups, fungi, and bacteria.

**Figure 5 foods-11-03945-f005:**
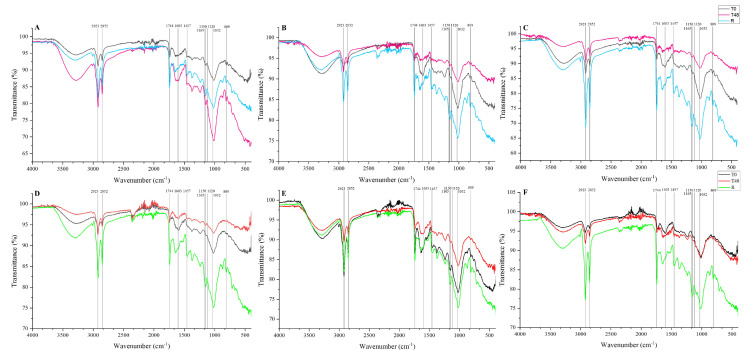
FTIR profile of each altitude and processing. N process: (**A**) (800 m), (**B**) (1200 m), and (**C**) (1400 m), PN process: (**D**) (800 m), (**E**) (1200 m), and (**F**) (1400 m).

**Figure 6 foods-11-03945-f006:**
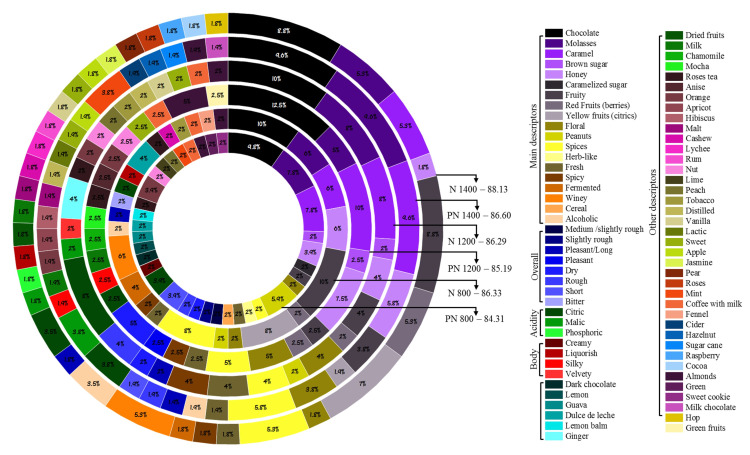
Sensory notes and descriptor occurrences were expressed in percentage for each altitude within each process.

## Data Availability

Data is contained within the article.

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
