# Peer review of "Molecular, Chemical, and Sensory Attributes Fingerprinting of Self-Induced Anaerobic Fermented Coffees from Different Altitudes and Processing Methods"

_foods, 2022, doi:10.3390/foods11243945_

Round 1

Reviewer 1 Report

The manuscript titled "Molecular, chemical, and sensorial attributes fingerprinting of self-induced anaerobic fermented coffees from different altitudes and processing methods" is well conducted and presented, however, some concerns need to be addressed. 

Sensory evaluation outcomes contribute greatly to the conclusion of such work, however, the number of candidates used in the sensory panel cannot be sufficient to make a substantial conclusion, unless a recommendation is provided. What about the consumer perspective? which statistical approach did you use to analyze the sensory evaluation data? the authors need to address these aspects appropriately.

The manuscript needs to be proofread. The authors should avoid starting a sentence with an abbreviation or a number. The chemical formulas of compounds need to be typed appropriately. 

The background information in the abstract is missing

The aim of the research in your introduction must be stated boldly instead of just mentioning the activities to be performed.

How were your samples handled to ascertain that the microbial population present did not receive any contamination from the equipment or the food handlers? furthermore, the drying and roasting conditions need to be specified. For exp. the temperature used and the fluctuations if any.

When assessing microbial presence during a fermentation process, you should clearly highlight how the presence of living cells was checked because a microorganism that occurred in the beginning and died before the end of fermentation will still have its DNA present at the end of fermentation. Clarify how you went about that aspect in your approach. 

When reporting the findings, the authors need to be consistent with the chosen format on how the values and their units are used and represented.

The authors need to ensure adequate visibility of the wording on the graphs. they are not very clear for some. 

Figure 1 should be labeled A, B as well 

The concluding section lacks recommendations for further studies

  Author Response

Round 2

Reviewer 2 Report

I strongly suggest the author double check the alkane series or ask an expert the standard procerdure of VOC identification by MS. The submitted supplementary ducument failed to show you have addressed the problem.

Author Response

My apologies, I did not understand what you meant in the previous comment. The alkane series that we used was from c7 to c40, which I already corrected in the manuscript, that is the series available in the market. Although the alkane series that we used was higher than some of our detected compounds, the program we use for compound analysis from Shimadzu does an automatic integration of the compounds below that series once is introduced in the program, that is how the data is treated and how is taught by experts. Moreover, to be certain that the volatiles that we detected are those compounds (since we know gc-ms is a sensitive technique) they must go through the first filter which is their mass spectra analysis after we have certainty that the mass spectra belong to that n compound they go through their second filter which is the calculation of their retention index based on the alkane series (this is not as reliable as analyzing their mass spectra) which the program does automatically. Finally, those compounds are searched in online chemical libraries, books, and scientific articles. If these filters are not done appropriately as stipulated by other studies, then perhaps are not addressed correctly.